# Does the Level of Air Pollution Affect the Incidence of Lung Adenocarcinoma in South-Eastern Poland?

**DOI:** 10.3390/ijerph20043177

**Published:** 2023-02-11

**Authors:** Marek Cierpiał-Wolan, Sebastian Wójcik, Jan Gawełko, Michalina Czarnota

**Affiliations:** 1Institute of Economics and Finance, College of Social Sciences, University of Rzeszów, 35-959 Rzeszów, Poland; 2Department, Statistical Office in Rzeszów, 35-959 Rzeszów, Poland; 3Institute of Mathematics, College of Natural Sciences, University of Rzeszów, 35-959 Rzeszów, Poland; 4Institute of Medical Sciences, College of Medical Sciences, University of Rzeszów, 35-959 Rzeszów, Poland; 5Institute of Health Sciences, College of Medical Sciences, University of Rzeszów, 35-959 Rzeszów, Poland

**Keywords:** air pollutants, morbidity, adenocarcinoma, lung cancer, Moran’s correlation, health impact

## Abstract

The aim of this study was to assess the association of long-term exposure to particulate matter with aerodynamic diameter, PM2.5, PM10, NO_2_ and SO_2_ as well as CO, with lung adenocarcinoma (AD) in south-east Poland for the years from 2004 to 2014. The study group consisted of 4296 patients with lung adenocarcinoma and the level of selected pollutants. To analyse the data, a standard statistics for cohort data, that is a risk ratio (RR), was used. The dependencies occurring in the distribution of pollutants and cancer incidence were examined using Moran’s I correlation coefficient. The current study suggests that PM10, NO_2_ and SO_2_ exposure as an air pollutant may increase female lung adenocarcinoma incidence. In men, the increased risk of adenocarcinoma lung cancer is affected by SO_2_ and PM10. A high morbidity rate in urban areas and suburbs may be connected with commuting from moderately polluted living areas to highly polluted working areas.

## 1. Introduction

Among the lung cancer risk factors, modified and non-modifiable factors can be distinguished. Cigarette smoking is an important risk factor in the aetiology of lung cancer. It has been well established throughout numerous epidemiological studies [1,2,3] and belongs to modifiable factors, such as passive smoking, and exposure to asbestos or radon. Among the non-modifiable factors, WHO indicates a personal or family history of lung cancer, previous lung radiotherapy and air pollution [4]. In the case of the latter, the risk is far less than the risk caused by smoking, but some researchers estimate that, worldwide, approximately five percent of all lung cancer deaths may be caused by outdoor air pollution [4]. There is scientific evidence of an increased incidence of respiratory, circulatory and cancer diseases due to exposure to various air pollutants, mainly to particles PM10, PM2.5 and their constituents [5,6,7,8] as well as to nitrogen dioxide (NO_2_) [9,10,11] or sulphur dioxide (SO_2_) [5,12]. Exogenous carbon monoxide (CO) is produced by the incomplete combustion of carbon-containing molecules. CO enters the circulatory system through the respiratory system, binding with haemoglobin in the blood, forming carboxyhaemoglobin (COHb); the affinity of haemoglobin for CO is much greater than for oxygen [13,14], which can cause hypoxia, despite the normal number of blood cells in the body [13,15]. Due to the fact that CO in the external environment is formed as a result of fuel combustion, it is one of the important elements of outdoor air pollution.

The most common types of lung cancer include squamous cell carcinoma (SCC), adenocarcinoma and small cell carcinoma (SCLC). Until the nineties of the twentieth century, SCC was the most common histologic subtype, particularly among men. Over the years, the situation has changed, causing an increase in the number of ADs and a simultaneous decrease in the number of cases of SCC. This trend has been observed in countries in North America, Europe and Asia [16,17]. However, the trend of the occurrence of lung cancer subtype depending on gender was maintained. Rates of AD relative to SCC and SCLC are greater in women [18]. As a result, the percentage of AD is increasing, and the incidence of lung cancer in women is also increasing.

Scientists from different parts of the world have conducted observations on the possible impact of long-term exposure to selected particles and gases of air pollutants on lung cancer and its variants [19,20,21,22,23,24,25]. Research is being conducted on the selective impact and the synergy effect between selected pollutants. Korean cohort studies observed a trend in the incidence of lung cancer, which has changed drastically in recent years. The first Korean study analysing the population of lung cancer patients, conducted in 1997, showed a significant prevalence of squamous cell carcinoma (44.7%) over adenoma (27.9%) [14]. In a re-examination ten years later, it turned out that the situation had changed drastically, as the majority of patients had the glandular type (48.8%). The second most common was squamous (27.2%). Studies have also shown differences in the incidence of lung cancer types depending on gender. The most common type of lung cancer in men is SCC and in women AD [26]. The increased affinity of the adenocarcinoma type in women has also been confirmed by other studies [27,28]. 

Most of the 50 most polluted cities in the European Union are located in Poland such as Cracow, Wroclaw or Warsaw [29]. Therefore, it is necessary to conduct observations and research on the levels of atmospheric air pollution and their harmful impact on public health, especially respiratory diseases.

The purpose of this project was to evaluate the effects of long-term exposure to outdoor air pollution PM2.5, PM10, NO_2_, and SO_2_ as well as CO at the place of residence on the morbidity of lung adenocarcinoma in south-east Poland.

Based on the aim of this work, the research questions were posed:(1)What is the incidence of adenocarcinoma of the lung in relation to gender?(2)Is there a relationship between the incidence of adenocarcinoma of the lung and the individual components of air pollution in the analysed area?(3)Is there a trend in the morbidity of adenocarcinoma cancer in a ten-year follow-up?

## 2. Materials and Methods

### 2.1. Lung Cancer and Pollution Data

To examine the relationship between long-term exposure to selected concentrations of air pollution at the place of residence and the incidence of lung adenocarcinoma, original pollution maps were created with the places of residence marked. The source of data on incidence was the report of morbidity of lung cancer from Clinical Voivodship Hospital number 1 in Rzeszów from 2004 to 2014. The number of all lung cancer cases registered in the indicated period was 10,993. Research group included 4296 patients with AD. The subjects were divided by gender, as in recent years there was a tendency to a higher incidence of AD in women [26,27,28,29,30]. The age was also divided. The source of data on SO_2_, NO_2_, PM10, PM2.5 and CO pollution was hourly data from pollution measuring stations located in the Podkarpackie Voivodship in 2005–2014 from the Voivodship Inspectorate for Environmental Protection. The lack of source data in small areas was supplemented based on the average annual values determined from the OS-1 report for a given area. An autoregressive model with an exogenous variable [28] was created for each station. The residences of patients with adenocarcinoma of the lung were superimposed on the spatial maps of air pollution created in this way. The results obtained made it possible to determine the level of influence of air pollutants considered selectively and their combinations on the incidence of lung adenocarcinoma of the province of lung in the Podkarpackie in the years 2004–2014. 

### 2.2. Data Analysis

The dependencies occurring in the distribution of pollutants and cancer incidence were examined using Moran’s I correlation coefficient [31]. This coefficient allows to assess the spatial autocorrelation, that is the relationship between the value of variable in a given grid cell in relation to the value of the variable in the neighbouring grid cells. The use of Moran’s I statistics is used for the identification of spatial clusters. Moran’s I correlation coefficient is given by
(1)$$I=\frac{n}{\sum_{i=1}^{n}w_{i}}\frac{\sum_{i=1}^{n}\sum_{j=1}^{n}w_{ij}\left(x_{i}-\bar{x}\right)\left(x_{j}-\bar{x}\right)}{\sum_{i=1}^{n}{\left(x_{i}-\bar{x}\right)}^{2}},$$
where: i,j=1,…,n—indices of spatial units; $\left[w_{ij}\right]$—row-standardized matrix of spatial weights; $x_{i}$—the variable of interest, $\bar{x}$—mean of $x_{\;}$.

If Moran’s I is close to one, then units tend to form spatial clusters. Opposite value can be interpreted as high local variability of variable of interest. No spatial pattern is indicated by Moran’s I close to zero.

Spatial analysis was carried out under assumption of Queen contiguity [32] (more neighbourhoods); that is, two cells are neighbours if they have at least one common point.

Finally, we used a risk ratio (RR), which is the ratio of the incidence rate in the exposed group and the incidence rate in the control group [27]. The exposed group are inhabitants of an area of above-average levels of particular pollutants, that is PM2.5, PM10, NO_2_, SO_2_ and CO, respectively. Confidence interval for RR was calculated and it was tested if RR was significantly different from 1 [33].

## 3. Results

The region has many urbanised areas in the central and western part. The southern part is low urbanised due to mountains. The eastern part is dominated by woods and is located close to the state border. The next choropleth confirms the general expectation that cancer incidences are correlated with the pollutant level. Nevertheless, there are areas where there are cancer incidences but quite low levels of pollutants and vice versa. Therefore we applied several tools to quantify this phenomenon. 

An analysis of the dependency between pollutants and cancer incidences was carried out in two ways: comparing the variables within a grid cell and comparing them with neighbouring grid cells. The places of residence of patients with lung adenocarcinoma were superimposed on the created spatial map of air pollution. The figure above shows the choropleth for NO_2_ (Figure 1). The map shows the area of the Podkarpackie Voivodeship, which is located in the south-eastern part of Poland, on a scale of 1:1,500,000. The area is divided into 21 counties and 4 cities. 

The table below shows Moran’s I correlation coefficient and the *p*-value of its significance test (Table 1).

For each analysed variable, there is a significant positive spatial autocorrelation but of low strength. Large grid clusters of low levels of pollutants in non-urban areas occur. Urban areas are very inhomogeneous with respect to the level of pollutants. Such a result is partially confirmed by the fact that the distribution of each pollutant reveals a high right-skewness and, in a case of cancer incidences, a very high right-skewness. Pollutants also reveal a much higher variability in urban areas in comparison to non-urban areas. The standard deviation of the pollutant level in urban areas is twice as much as in non-urban areas. Moreover, in the first case, the distribution has a heavy right tail (Figure 2).

The study area was divided into below-average and above-average areas of pollution, respectively, for each pollution. In the next step, the people living in these areas were evaluated and broken down by age (below 75 and older) as well as by gender. The division into such age groups resulted from the median age of the people in the cancer incidence database. 

In the group of patients under 75 years of age, the impact of air pollution on the incidence of AD of the lung was observed to a greater extent in women, especially in relation to SO_2_, NO_2_ and PM10. In contrast to women, no exposure to NO_2_ was observed in men (Table 2). 

The analysis showed that the older group is exposed to the carcinogenic effects of all the analysed air pollutants, regardless of gender (Table 3).

## 4. Discussion

In 2013 the International Agency for Research on Cancer classified outdoor air pollution as carcinogenic to humans [34]. Worldwide, it has been estimated that 6% of all lung cancer deaths are attributable to ambient air pollution [35]. 

The subject of research all over the world is the analysis and attempt to determine the impact of long-term exposure to particulate matter in the air on the incidence of lung cancer [19,20,21,22,23,24,25]. The available results are not conclusive. Long-term observations of the impact of exposure to air pollution and their assessment as a carcinogen increase the value of research. This study assessed the relationship between long-term exposure to various air pollutants in the place of residence and the occurrence of lung adenocarcinoma in adult Poles in south-eastern Poland. 

An analysis of pollutants was carried out in two ways: comparing within a grid cell and comparing with neighbouring grid cells with an accuracy of 1 km^2^. Several patterns of spatial distribution of pollutants were discovered with the use of the pollution model, standard and spatial correlation coefficient, and density estimation. There are large areas with a stable low level of all pollutants, mostly rural areas. Cancer incidences seem to not be connected with pollutants. Next there are urban areas with very unstable and high levels of pollutants, which quite quickly decreases the levels of pollutants in the suburbs. The third type are industrial areas with industries that spread pollution over a large neighbourhood. The impact of these pollutant sources on cancer incidences is visible in neighbouring rural areas. New industrial zones are being created in the study area. These observations and further research may influence the creation of a certain border zone separating rural areas adjacent to industrial areas, creating a potential protective zone, not inhabited by society. 

AD is the most common histological type of lung cancer in the population [15], with higher rates of AD occurring in women, 189 cases [14,15]. Despite other factors influencing the disease, such as lifestyle, diet or climate change, a trend of increased incidence among women was also observed in the study group.

In the European Study of Cohorts for Air Pollution Effects, data were used from 17 cohort studies based in nine European countries, and the meta-analyses showed a statistically significant association between higher hazards of risk for lung cancer with higher exposures to PM2.5 and PM10. Higher results of PM2.5 and PM10 were stronger associated specifically with AD [36]. These studies confirmed the carcinogenic effect of PM10 particulate matter in the case of all respondents and PM2.5 only in the case of people over 75 years of age (Table 2 and Table 3). 

Slightly different results were shown in the seven cohorts from Europe. The analysis showed that long-term exposure to PM2.5 in the place of residence can affect the growth of lung cancer morbidity [30]. These analyses verified the ten-year impact of exposure to selected air pollutants but did not show such a strong impact on the increase in the incidence of AD of the lung in the case of PM2.5. In the case of PM 2.5 for the cohort of men under the age of 75, the risk ratio was very close to 1; hence, it was not interpreted.

The contribution of traffic emissions to ambient air pollution is assessed as individual traffic-related pollutants such as NO_2_, PM2.5 and benzene. The NO_2_ level is often used as a marker for traffic-based air pollution [37,38]. In the European Study of Cohorts for Air Pollution Effects, the relationship between exposure to NO_2_ and the occurrence of lung cancer was not confirmed [36]. However, a more recent meta-analysis of 20 studies from different parts of the world reported consistent evidence for the association between NO_2_ and lung cancer. In Europe, there was a relative risk between exposure to NO_2_ and the morbidity of lung cancer [39]. Our previous work showed nitrogen dioxide to be a lung cancer carcinogen. Once NO_2_ is mixed with the other outdoor air pollution, it becomes more persistent in terms of explaining the appearance of cancers and could constitute the main cause of SCC [11]. For the cohort of men under the age of 75, the risk ratio was less than 1 in the case of NO_2_. Thus, an above-average level of NO_2_ did not lead to the higher risk of lung cancer. For all other pollutants and cohorts, the risk ratio was always greater than 1, indicating a higher risk of lung cancer for areas of above-average levels of pollutants. 

The American Cancer Society Study and the Adventist Health Study indicate a strong relationship between SO_2_ exposure and increased lung cancer mortality [40,41,42]. This association was confirmed by studies conducted among non-smokers in California [42]. The results showed that lung cancer incidence was positively associated with interquartile range increases for SO_2_ in women. A particularly significant effect of SO_2_ on the incidence of lung adenocarcinoma in women was also confirmed by this study (Table 2 and Table 3).

Summing up the conducted causal analyses, it is worth emphasizing that scientific reports with such precise geolocation are rarely encountered, which increases the credibility of linking morbidity with its factors.

## 5. Conclusions

The provided scientific evidence may affect the health of the study population in the future in two ways. In terms of adenocarcinoma prophylaxis: by recommending more frequent check-ups in the risk areas of the Podkarpackie Voivodship with the highest concentrations of PM10, NO_2_ and SO_2_, and in particular in rural areas in close proximity to an industrial zone, or by recommending a change in residence due to a confirmed carcinogenic effect of the external environment. Thanks to the presented data and the created models enabling the identification of areas with the highest concentrations of selected pollutants, we can now identify those people most at risk of developing lung cancer, broken down by histological subtypes, gender and age.

The declining tendency in smoking points to the necessity of focusing on other risk factors. The analysis of these within the context of morbidity and mortality can help to develop more effective screening programs. The present study found significant associations between ambient air pollution and lung cancer.

The current study suggests that PM10, NO_2_ and SO_2_ exposure as air pollutants may increase female lung adenocarcinoma incidence. In men, the increased risk of adenocarcinoma lung cancer is affected by SO_2_ and PM10. 

A high morbidity rate in urban areas and suburbs may be connected with commuting from moderately polluted living areas to highly polluted working areas.

## Figures and Tables

**Figure 1 ijerph-20-03177-f001:**
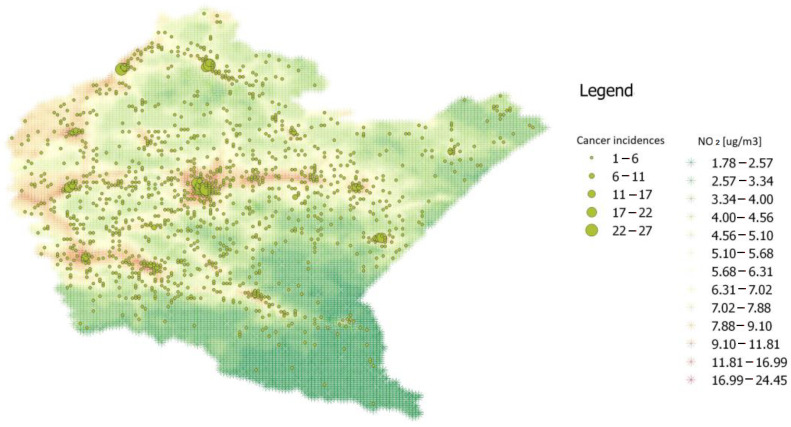
Choropleth of cancer incidences and NO_2_ pollution levels.

**Figure 2 ijerph-20-03177-f002:**
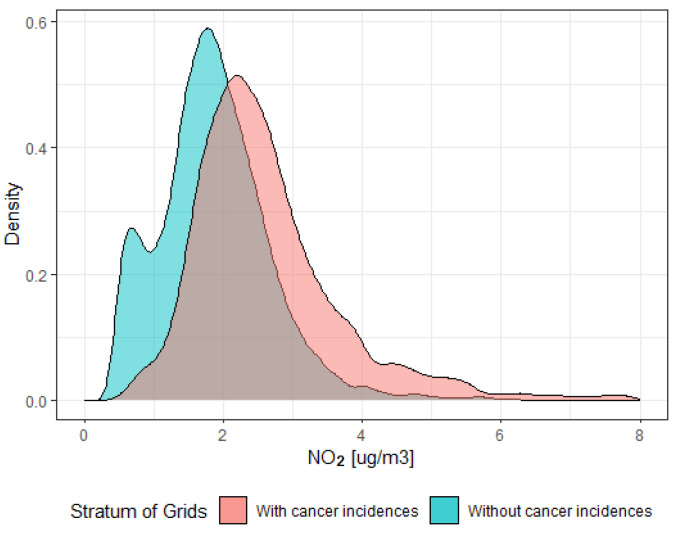
Distribution of NO_2_ pollution level.

**Table 1 ijerph-20-03177-t001:** Moran I statistics for air pollution and cancer incidences.

	Cancer Incidences	SO_2_	NO_2_	PM2.5	PM10	CO
Moran I statistics	0.0197	0.3064	0.2406	0.192	0.2173	0.2214
Standard deviation of Moran’s I	5.34	82.0264	64.4287	51.4049	58.1762	58.474
*p*-value	<0.0001	<0.0001	<0.0001	<0.0001	<0.0001	<0.0001

**Table 2 ijerph-20-03177-t002:** Risk ratio for the group below 75 years of age.

Risk Ratio					
	SO_2_	NO_2_	PM2.5	PM10	CO
Women	1.203 (1.036–1.396)	1.053 (1.006–1.418)	0.974 (1.053–1.325)	1.013 (1.072–1.380)	0.938 (0.939–1.265)
*p*-value	0.001	0.000	0.003	0.010	0.054
Men	1.092 (0.986–1.209)	0.984 (0.877–1.208)	0.999 (0.899–1.227)	1.040 (0.918–1.278)	0.966 (0.928–1.179)
*p*-value	0.026	0.032	0.017	0.003	0.042
Total	1.121 (1.030–1.219)	1.034 (0.923–1.224)	1.020 (0.949–1.210)	1.061 (0.968–1.260)	0.986 (0.969–1.163)
*p*-value	0.001	0.001	0.004	0.000	0.013

**Table 3 ijerph-20-03177-t003:** Risk ratio for the group over 75 years of age.

Risk Ratio					
	SO_2_	NO_2_	PM2.5	PM10	CO
Women	1.146 (0.955–1.376)	1.104 (0.917–1.330)	1.223 (1.020–1.466)	1.188 (0.986–1.430)	1.122 (0.939–1.341)
*p*-value	0.040	0.119	0.005	0.027	0.035
Men	1.046 (0.928–1.178)	1.015 (0.899–1.445)	1.043 (0.925–1.177)	1.038 (0.918–1.174)	1.041 (0.927–1.169)
*p*-value	0.203	0.400	0.234	0.278	0.180
Total	1.081 (0.978–1.194)	1.047 (0.946–1.159)	1.101 (0.995–1.217)	1.086 (0.980–1.204)	1.068 (0.969–1.177)
*p*-value	0.040	0.167	0.021	0.057	0.036

## Data Availability

Not applicable.

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
