# Peer review of "Does the Level of Air Pollution Affect the Incidence of Lung Adenocarcinoma in South-Eastern Poland?"

_ijerph, 2023, doi:10.3390/ijerph20043177_

Round 1

Reviewer 1 Report

„Does the level of air pollution affect the incidence of lung adenocarcinoma in South-Eastern Poland?” focuses on the association of long-term exposure to outdoor air pollution with lung adenocarcinoma (AD) in South East Poland from the years 2004 to 2014. The Authors have explored studies concerning adenocarcinoma, squamous cell carcinoma and small cell carcinoma. The target group consisted of 4296 patients and they were divided by gender and age. The study was conducted properly, however I have some special comments.

11.      In lines 52-54 on 2nd page the Authors have observed the greater tendency of lung cancer in women. What is the cause?

22.      Line 68 on 2nd page. I would suggest that Authors consider name few of the cities. Moreover, the industries which affect the pollution may be also added.

33.      To the Discussion section, the Authors have omitted one major issue, mainly occupational expore. Some of the workers work outside thus they are exposed to the outdoor air pollution longer than 8 hours a day.

Author Response

Dear Reviewer,

Thank you for your time. In reference to your suggestions, I’m sending replies to the left comments. I believe that with these changes, the article will strengthen the argument of the article scientifically and will be a source of knowledge for other readers.

Comment 1.In lines 52-54 on 2nd page the Authors have observed the greater tendency of lung cancer in women. What is the cause?

Replay: The trend of increased incidence of adenocarcinoma of the lung among women was observed in American studies. Analysis of US lung cancer incidence patterns overall and by histologic type and cigarette smoking that spans a century of birth cohorts has confirmed a number of previous observations as well as identifying several new observations. Overall rates have been declining for several decades among males of each racial/ethnic group; rates among females have plateaued. Adenocarcinoma rates are continuing to climb among females while generally declining among males. All lung carcinoma histologies are associated with smoking but with more modest for adenocarcinoma. These trends may have occurred owing to changes in smoking behaviour and cigarette composition.

The reason for the change in the incidence trend in young women is not fully known. The most frequently indicated main factor of morbidity, which is smoking, has no basis in the case of people who have never smoked. Therefore, other carcinogenic factors that may affect morbidity and greater predisposition depending on sex are sought.

Other environmental exposures associated with lung cancer include arsenic, asbestos, and radon, as well as outdoor air pollution which are known carcinogens.

This study is an attempt to assess the long-term exposure to air pollution on the incidence of adenocarcinoma of the lung with gender specific.

Comment 2. Line 68 on 2nd page. I would suggest that Authors consider name few of the cities. Moreover, the industries which affect the pollution may be also added.

Replay: We made corrections to the text

Comment 3: To the Discussion section, the Authors have omitted one major issue, mainly occupational expore. Some of the workers work outside thus they are exposed to the outdoor air pollution longer than 8 hours a day.

Replay: A valid reviewer's suggestion that has been added to the discussion. Thank you for your valuable tip.

Reviewer 2 Report

The issue of this manuscript is very interesting, while the following commented should be addressed before published in this journal.

1.     Line 38-29: References are too much. Please just cite the relevant ones.

2.     Figure 1: It had better to give more information of the map, such as scales, administrative areas and names, because many readers may not familiar with the map.

3.     Could the author give and explanation about why NO2 was not relevant to the risk of adenocarcinoma lung cancer of men.

4.     Discussion: The current discussion is too superficial and not very relevant to the thesis. The author should conduct an in-depth discussion based on the results.

5.     The format of the references is inconsistent, please organize them according to the requirements of the journal.

Author Response

Dear Reviewer,

Thank you for your time. In reference to your suggestions, I’m sending replies to the left comments. I believe that with these changes, the article will strengthen the argument of the article scientifically and will be a source of knowledge for other readers.

  1. Line 38-29: References are too much. Please just cite the relevant ones.

We made corrections to the text

  1. Figure 1: It had better to give more information of the map, such as scales, administrative areas and names, because many readers may not familiar with the map.

We made corrections to the text

  1. Could the author give and explanation about why NO2 was not relevant to the risk of adenocarcinoma lung cancer of men.

For the cohort of men under the age of 75, Risk Ratio is less than 1 in a case of NO2. Thus, above-average level of NO2 did not lead to the higher risk of lung cancer. For all other pollutants and cohorts, Risk Ratio is always greater than 1, indicating higher risk of lung cancer for areas of  above-average level of pollutants. In a case of PM 2.5 for the cohort of men under the age of 75, Risk Ratio is very close to 1, hence it is not interpreted.

Additional explanations were included in the article for clarifications.

  1. Discussion: The current discussion is too superficial and not very relevant to the thesis. The author should conduct an in-depth discussion based on the results.

We made corrections to the text

  1. The format of the references is inconsistent, please organize them according to the requirements of the journal.

We made corrections to the text

With Kind Regards,

Michalina Czarnota